METHODS AND RESOURCES

# CellTracksColab is a platform that enables compilation, analysis, and exploration of cell tracking data

Estibaliz Gómez-de-Mariscal[1], Hanna Grobe[2], Joanna W. Pylvänäinen[2,3],
Laura Xénard[4,5], Ricardo Henriques[1,6], Jean-Yves Tinevez[4],
Guillaume Jacquemet [2,3,7,8]*

1 Instituto Gulbenkian de Ciência, Oeiras, Portugal, 2 Faculty of Science and Engineering, Cell Biology, Åbo Akademi University, Turku, Finland, 3 InFLAMES Research Flagship Center, University of Turku and Åbo Akademi University, Turku, Finland, 4 Institut Pasteur, Université Paris Cité, Image Analysis Hub, Paris, France, 5 Institut Pasteur, Université Paris Cité, INSERM UMR1225, Pathogenesis of Vascular Infections, Paris, France, 6 UCL Laboratory for Molecular Cell Biology, University College London, London, United Kingdom, 7 Turku Bioscience Centre, University of Turku and Åbo Akademi University, Turku, Finland, 8 Turku Bioimaging, University of Turku and Åbo Akademi University, Turku, Finland

☉ These authors contributed equally to this work.
* guillaume.jacquemet@abo.fi

**Data Availability Statement:** Multiple test datasets are available on Zenodo. They include the two test datasets that can be directly downloaded from within the CellTracksColab notebooks (8413510,

## Abstract

In life sciences, tracking objects from movies enables researchers to quantify the behavior of single particles, organelles, bacteria, cells, and even whole animals. While numerous tools now allow automated tracking from video, a significant challenge persists in compiling, analyzing, and exploring the large datasets generated by these approaches. Here, we introduce CellTracksColab, a platform tailored to simplify the exploration and analysis of cell tracking data. CellTracksColab facilitates the compiling and analysis of results across multiple fields of view, conditions, and repeats, ensuring a holistic dataset overview. CellTracksColab also harnesses the power of high-dimensional data reduction and clustering, enabling researchers to identify distinct behavioral patterns and trends without bias. Finally, CellTracksColab also includes specialized analysis modules enabling spatial analyses (clustering, proximity to specific regions of interest). We demonstrate CellTracksColab capabilities with 3 use cases, including T cells and cancer cell migration, as well as filopodia dynamics. CellTracksColab is available for the broader scientific community at https://github.com/CellMigrationLab/CellTracksColab.

## Introduction

In life science, tracking has emerged as an indispensable tool for unparalleled insights into dynamic molecular and cellular behaviors. Parallel to this, segmentation methods relying on machine learning and deep learning are now greatly facilitating the implementation of complex tracking pipelines [1–4], enabling the quantitative analysis of these dynamic behaviors.

8420011). In addition, the three datasets showcased in this study, their tracking files, and the CellTracksColab results are also available on Zenodo (11282716, 11286110, 11285514). The code for CellTracksColab is publicly available under the MIT license, encouraging broad utilization and adaptation. CellTracksColab's GitHub repository serves as a dynamic platform for tracking the evolution of the code across various versions. Users are encouraged to report issues and suggest features directly through the GitHub interface. A stable version of the code and associated documentation is also archived on Zenodo (11384844).

**Funding:** This study was supported by the Research Council of Finland (338537 to GJ, https://www.aka.fi/en/), the Sigrid Jusélius Foundation (to GJ, https://www.sigridjuselius.fi/en/), the Cancer Society of Finland (Syöpäjärjestöt; to GJ, https://www.cancersociety.fi/), and the Solutions for Health strategic funding to Åbo Akademi University (to GJ, https://www.abo.fi/en/solutions-for-health/). This research was supported by the InFLAMES Flagship Programme of the Academy of Finland (decision numbers: 337530, 337531, 357910, and 357911, https://www.aka.fi/en/). EGM and RH received funding from the European Union through the Horizon Europe program (AI4LIFE project with grant agreement 101057970-AI4LIFE, and RT-SuperES project with grant agreement 101099654-RTSuperES to RH, https://research-and-innovation.ec.europa.eu/funding/funding-opportunities/funding-programmes-and-open-calls/horizon-europe_en). EGM and RH also acknowledge the support of the Gulbenkian Foundation (Fundação Calouste Gulbenkian, https://gulbenkian.pt/en/) and the European Research Council (ERC) under the European Union's Horizon 2020 research and innovation program (grant agreement No. 101001332 to RH, https://erc.europa.eu/homepage). LX received funding from the INCEPTION project (PIA/ANR-16-CONV-0005, https://anr.fr/) and is a student from the FIRE PhD program funded by the Bettencourt Schueller Foundation (https://www.fondationbs.org/en) and the EURIP graduate program (ANR-17-EURE-0012, https://www.learningplanetinstitute.org/en/eurip-graduate-school/). This study was also supported by France BioImaging (Investissement d'Avenir; ANR-10-INBS-04, J.-Y. T., LX, https://france-bioimaging.org/). This work was also supported by the European Molecular Biology Organization (EMBO, https://www.embo.org/) Installation Grant (EMBO-2020-IG-4734 to RH), the EMBO Postdoctoral Fellowship (EMBO ALTF 174-2022 to EGM), the Chan Zuckerberg Initiative (https://chanzuckerberg.com/) Visual Proteomics

Yet, as the capabilities of tracking tools have expanded, so too have the challenges associated with analyzing the resulting data.

Multiple tools have been developed to help researchers compile tracking data; for instance, these include the Ibidi Chemotaxis tool (a Fiji plugin), the MotilityLab website (an online platform for CelltrackR [5]), or TrackMateR [6]. These traditional analytical approaches, implemented by us and many others, typically reduce tracking datasets to population-level analyses where track metrics are averaged across different conditions. Yet, while practical, such analyses overlook the heterogeneity within biological data. Over the past 2 years, multiple tools, including CellPhe (an R toolbox [7]), Traject3D (a collection of MATLAB scripts [8]), and CellPlato (a Python toolbox [9]), have been designed to harness the high dimensionality of tracking datasets to assist in the unbiased discovery of rare phenotypes. Still, these tools often remain difficult to implement for users with no or little coding expertise.

Here, we present CellTracksColab, a Python-based platform to streamline the analysis of tracking datasets. This platform is specifically designed for researchers, particularly those with limited programming expertise, facilitating the exploration and analysis of tracking data. CellTracksColab leverages the power of Jupyter notebooks, which blend live code execution with comprehensive documentation access. CellTracksColab can run locally and in the cloud, accommodating diverse user preferences and resource availability. Drawing on successful models like ColabFold [10] and ZeroCostDL4Mic [11], CellTracksColab is fully integrated within the Google Colaboratory framework (Colab). Through a simplified workflow, researchers can install essential software dependencies with a few mouse clicks, upload their tracking data, and run their analyses. CellTracksColab extends beyond visualization and population analyses, empowering researchers to delve into the nuanced dynamics and behaviors encapsulated within their tracking experiments. We first describe CellTracksColab's architecture. Then, we demonstrate CellTracksColab features and capabilities in studying T cells and cancer cell migration and filopodia dynamics.

## Results

### The CellTracksColab framework

The CellTracksColab platform comprises a collection of Jupyter notebooks designed to streamline tracking data analysis (Fig 1A). CellTracksColab can be run locally or in cloud services such as Google Colab, which provides users free access to computing resources that simplify the user experience by eliminating the need for local installations.

CellTracksColab is designed to process tracking data from various open-source tracking software, including TrackMate [1], CellProfiler [12], Icy [13], ilastik [14], and the Fiji Manual Tracker [15]. CellTracksColab supports tracking data stored in XML (TrackMate) and CSV formats (TrackMate, CellProfiler, Icy, ilastik, and Fiji Manual Tracker). CellTracksColab can also be made compatible with other tracking tools that export results that follow our minimal requirements (see documentation for details). To facilitate a structured analysis, users are advised to organize their files into directories representing different experimental conditions and biological repeats. This organizational strategy is crucial for accurately categorizing and analyzing the dataset, considering various aspects such as experimental conditions, biological replicates, and fields of view. By promoting structured data management, CellTracksColab streamlines the analytical process and enhances the exploration and understanding of data variability and heterogeneity across the dataset.

The performance of CellTracksColab is limited by the resources available to the user, particularly the amount of RAM available, which can limit the volume of data that can be processed. However, optimization of the underlying code has been executed to ensure maximal efficiency

Grant (vpi-0000000044 with DOI:10.37921/743590vtudfp to RH). RH also acknowledges the support of LS4FUTURE Associated Laboratory (LA/P/0087/2020, https://www.ls4future.pt/). The open access publication fees were funded by the Gösta Branders research fund, Åbo Akademi Research Foundation (Gösta Branders forskningsfond, Stiftelsen för Åbo Akademi, https://stiftelsenabo.fi/en/home/). While the European Union funded this study, the views and opinions expressed are those of the authors only and do not necessarily reflect those of the European Union. Neither the European Union nor the granting authority can be held responsible. None of the funders listed above were involved in the design and execution of this study.

**Competing interests:** The authors have declared that no competing interests exist.

**Abbreviations:** FAIR, Findable, Accessible, Interoperable, and Reusable; FOV, field of view; HDBSCAN, Hierarchical Density-Based Spatial Clustering of Applications with Noise; MAE, mean absolute error; R3, third biological repeat; t-SNE, t-distributed Stochastic Neighbor Embedding; UMAP, Uniform Manifold Approximation and Projection.

in resource utilization. For instance, we analyzed more than 50,000 tracks (>3 million objects from 117 videos) using CellTracksColab and the free version of Google Colab (all results presented in the manuscript can be replicated with the free version of Google Colab). CellTracksColab could accommodate one of our larger datasets encompassing over 536,000 tracks (>56 million objects from 300 videos), but this required the additional RAM that Google Colab Plus provides.

## Analyzing data using CellTracksColab

When the tracking data are loaded into CellTracksColab, these are automatically exported into the CellTracksColab format. This standardized format ensures consistent data access and manipulation, facilitating thorough analysis of the tracking data across the platform. Once exported, users can, for example, visualize (Fig 1B), filter, and smooth tracks (S1 Fig). Track smoothing using moving averages can prove to be particularly beneficial before the computation of directionality metrics, especially when the tracked object exhibits jitteriness (for instance, nuclei) and the user's interest lies in discerning the overall movement of the cell.

Upon loading data, CellTracksColab can compute various track metrics or import them directly from prior analyses conducted in the tracking software (Fig 1A). This is ensured by the flexible design of the CellTracksColab format, which provides the aggregation of additional metrics without affecting the content of the original dataset. Users can then generate boxplots illustrating the distribution of different track metrics of interest (Fig 1C). Additionally, several relevant statistical metrics are calculated, such as Cohen's *d* value—which quantifies the standardized effect size between groups and is less sensitive to sample size variations—and the *p*-values of statistical hypothesis tests—which compare the distribution of track metrics across conditions. The statistical tests available include a randomization test that assesses the distribution of Cohen's *d* values obtained with bootstrapping and *t* tests that compare the mean value distributions obtained from bootstrapping, following the SuperPlots methodology [16]. Both tests are available with and without Bonferroni correction, which adjusts the *p*-value to account for multiple comparisons (Fig 1C). CellTracksColab also enables users to perform quality control on their dataset, such as checking that their data are balanced between repeats and conditions. Namely, the user can resample unbalanced data before plotting the track metrics of interest. In addition, CellTracksColab can compute similarities across different experimental conditions and replicates using various track metrics to ensure data reliability and meaningful analysis. The results are visualized using hierarchical clustering in the form of dendrograms, which aids in comparing similarities within and across different conditions and identifying outliers.

Inspired by CellPlato [9], CellTracksColab integrates Uniform Manifold Approximation and Projection (UMAP) or t-distributed Stochastic Neighbor Embedding (t-SNE) combined with Hierarchical Density-Based Spatial Clustering of Applications with Noise (HDBSCAN) to explore the inherent heterogeneity within tracking datasets unbiasedly (Fig 1D). This combination allows for dimensionality reduction and effective clustering of tracks. The platform provides capabilities to plot track metrics for each cluster, creates heatmaps for an overview of data variability, and identifies exemplar tracks, representatives of each cluster, for detailed analysis. CellTracksColab also includes specialized spatial analysis modules that enable the spatial analysis of track data. These modules enable, for instance, the assessment of track clustering (Fig 1E) or calculating the proximity of tracks to specific regions of interest (Fig 1F). These tools facilitate the discovery of distinct subpopulations or behaviors within the data and also serve dual purposes: identifying actual clusters and categorizing data for comparative fingerprinting.

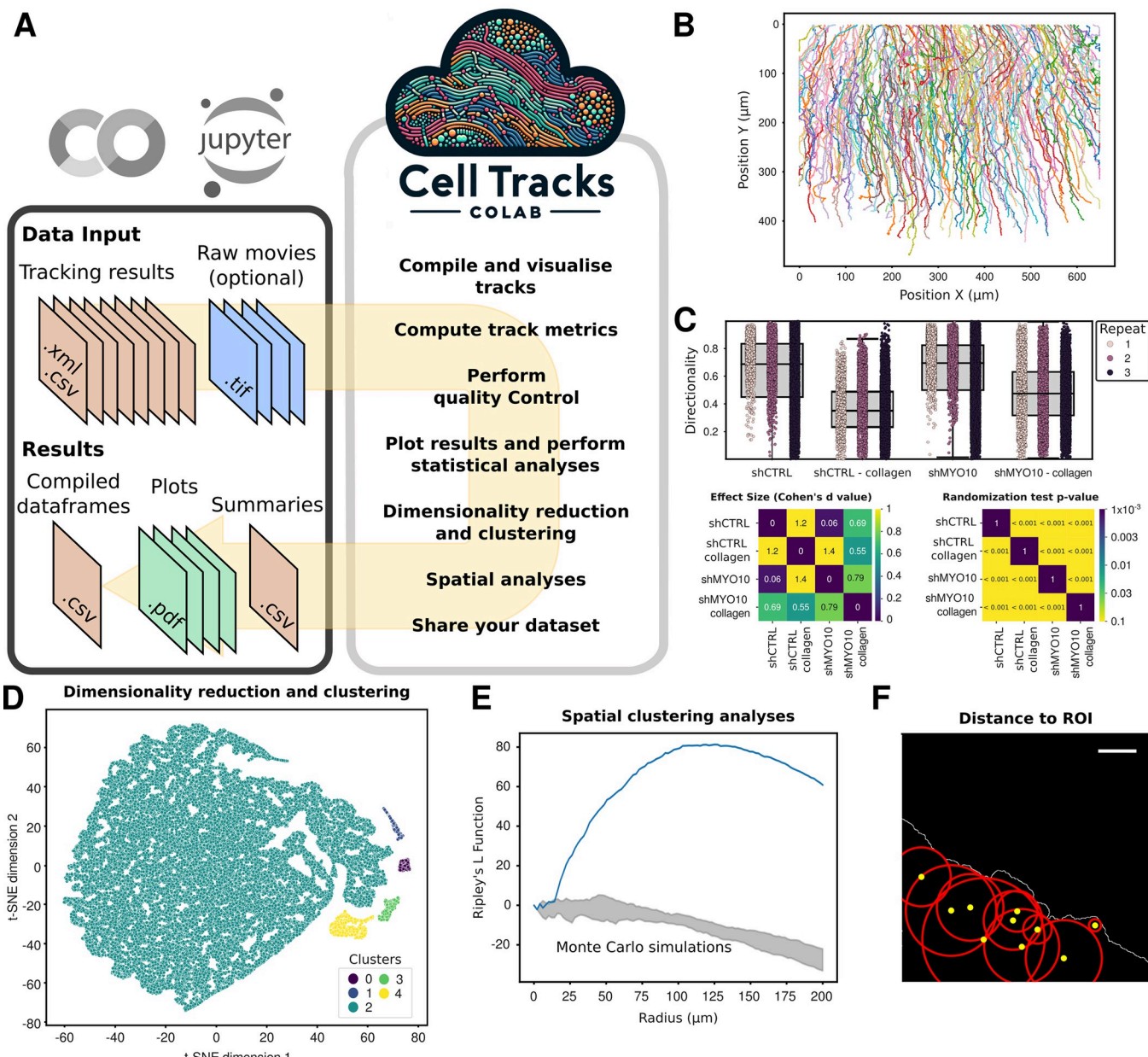

**Fig 1. The CellTracksColab platform.** (**A**) Schematic representation of the CellTracksColab workflow. (**B**) Visualization of tracks in a CellTracksColab notebook. (**C**) Statistical analysis of track metrics using CellTracksColab. This figure shows the analysis of breast cancer cell migration (expressing CTRL shRNA or MYO10-targeting shRNA) in different environments beneath a collagen gel and standard media. The directionality metric is presented in a Tukey boxplot format. Vertical whiskers extend to data points within 1.5× the interquartile range. Each biological replicate is uniquely color-coded for clarity. Accompanying the plot are mirrored heatmaps that illustrate the effect size (Cohen's *d* value) and statistical significance (*p*-values from randomization tests) across various conditions. Underlying numerical data can be found in S1 Data. (**D**) Dimensionality reduction and clustering visualization in CellTracksColab. This panel displays a 2D t-SNE projection of the entire dataset, utilizing comprehensive track metrics for the analysis. Data points are color-coded to reflect cluster groups identified through HDBSCAN analysis on the t-SNE projection, providing insights into track characteristics and similarities. (**E**) Spatial clustering analysis using Ripley's L function and Monte Carlo simulations in CellTracksColab. This graph illustrates the spatial distribution of tracks, where a blue curve above the zero line indicates clustering at a specific radius in the field of view. The Monte Carlo simulation results are included to assess the statistical significance of the observed patterns. (**F**) Measurement and analysis of object-to-region proximity using CellTracksColab. This example demonstrates the platform's utility in quantifying the distance of objects (marked as yellow dots) relative to a defined region of interest (denoted by the white edge). The tool allows tracking these distances over time and computing related metrics, facilitating in-depth spatial analysis.

Importantly, PDF files of all plots and CSV files encapsulating all plot data are exported, enabling users to visualize and revisit the results using their preferred software platforms. Due to the platform's flexible design, we anticipate the addition of new analysis modules, both by our team and the user community.

### Exploring T cell migration using CellTracksColab

To showcase the capabilities of CellTracksColab, we first chose to reanalyze a small dataset of T cells migrating on either vascular cell adhesion protein 1 (VCAM) or intercellular adhesion molecule 1 (ICAM), captured through brightfield microscopy (Fig 2A) [1,17,18]. Automated cell tracking was achieved using StarDist and TrackMate algorithms [1]. The dataset encompasses 10 videos spread across 2 conditions and 3 biological repeats. CellTracksColab compiled this dataset using Colab in a few seconds, incorporating 2,297 tracks and 38,852 tracked objects.

After the computation of additional metrics, we first evaluated the dataset's balance and variability across different fields of view. Although the dataset exhibited some condition imbalance, we opted against resampling due to its relatively small size (S2A Fig). Intriguingly, the field of view (FOV)-based clustering analysis unveiled an unexpected alignment between 2 FOVs from the ICAM condition with those from the VCAM condition, hinting at potential similarities in cell tracking patterns (S2B Fig). The condition and repeat-based clustering analysis further corroborated this observation. Specifically, the analysis revealed that the ICAM second biological repeat displayed a clustering pattern remarkably similar to those observed in VCAM repeats (S2C Fig). This analysis indicates that this particular ICAM biological repeat does not behave as the other two, providing valuable information.

We further utilized CellTracksColab to plot key track metrics, including mean speed and directionality of T cell migration. Our analysis confirmed that T cells exhibit slower and less directional movement on VCAM than on ICAM surfaces (Fig 2B). To delve deeper, we employed UMAP for dimensionality reduction, followed by HDBSCAN clustering. This approach revealed the presence of at least 5 distinct behavioral clusters within the cell population, suggesting varied migration patterns among T cells (Fig 2C and 2D). A fingerprinting plot then provided insights into the distribution of ICAM and VCAM tracks across these clusters, highlighting differing proportions (Fig 2E). A notable observation was the much higher percentage of ICAM tracks in cluster 3 compared to VCAM and a higher percentage of VCAM tracks in cluster 1 compared to ICAM. CellTracksColab generates a heatmap representing the Z-score of available track metrics for each cluster to facilitate rapid metric comparison across clusters (Fig 2F). Cluster 3 comprises fast and more directional tracks. In contrast, cluster 1 primarily comprises very slow-migrating cells (Fig 2G). Finally, we compared track metrics between the ICAM and VCAM conditions within specific clusters. Focusing solely on tracks in cluster 4 (a cluster composed of migrating cells), we observed that among motile cells, cells plated on ICAM migrated faster and tended to be more circular than those on VCAM (Fig 2H). While we provide only brief examples here, we can delve deeper into the analysis, identify tracks belonging to each cluster, and match them back to the original video. Further analyses will depend on the user's interest in the biological phenomenon studied. This multifaceted analysis underscores CellTracksColab's utility in offering nuanced insights into cell migration dynamics under different conditions.

### Studying cancer cell migration using CellTracksColab

Next, we analyzed a new dataset of collectively migrating cancer cells (Fig 3A). In this dataset, breast cancer cells expressing a CTRL shRNA or a shRNA targeting the filopodial protein

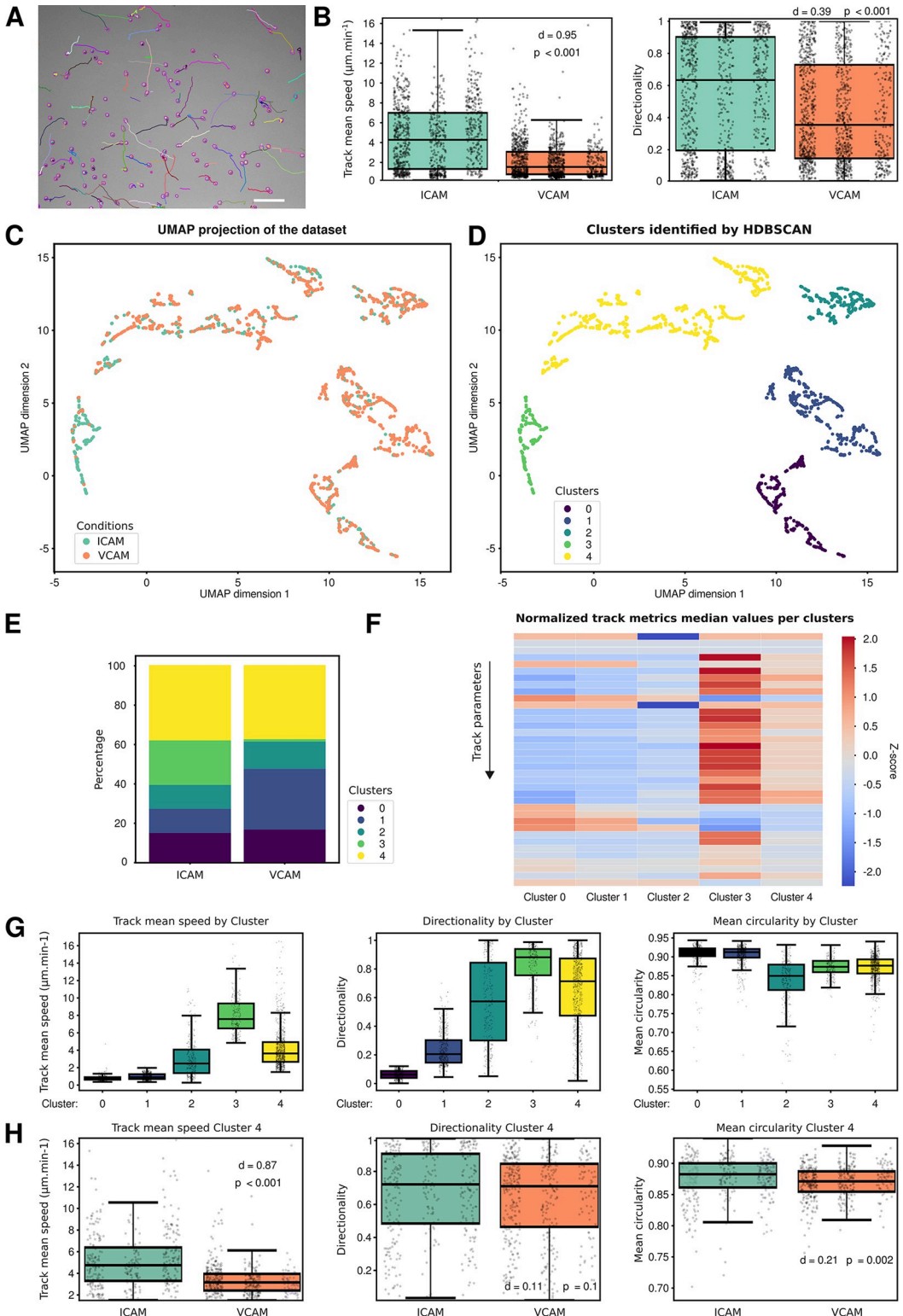

**Fig 2. Exploring T cell migration using CellTracksColab.** (**A**) T cells plated on ICAM were recorded using a brightfield microscope and automatically tracked using StarDist and TrackMate. Detected cells (in magenta) and their tracks (colors indicate track ID) are displayed. Scale bar: 100 μm. (**B**) The "track mean speed" and track "directionality" metrics for each condition are summarized in Tukey boxplots. The effect size (d, Cohen's *d* value) and the statistical significance (p, *p*-values from randomization tests) between the conditions are displayed. (**C**) 2D UMAP projection of the entire dataset. Data points

are color-coded based on VCAM and ICAM conditions. (**D**) Resultant clusters from the HDBSCAN analysis on the 2D UMAP projection. Euclidean distance served as the metric for clustering. Each identified cluster is color-coded. (**E**) Fingerprint plot showcasing the distribution percentage of track in each cluster across different conditions. (**F**) Heatmap representation, normalized using Z-scores, displaying variations in selected track metrics among the clusters. Full heatmaps are available in the Zenodo archive of this dataset. (**G**) The "track mean speed," track "directionality," and "mean (cell) circularity" metrics for each cluster are summarized in a Tukey boxplot format as in (B). (**H**) The "track mean speed," track "directionality," and "mean (cell) circularity" metrics for each condition for cluster 4 are summarized in a Tukey boxplot format as in (B). For all box plots, the vertical whiskers extend to data points within 1.5× the interquartile range, and the values for each track are shown as dots. Each biological replicate is displayed next to each other from R1 to R3 (left to right). Plot axes are limited to 10× the interquartile range. (B, G, and H) Underlying numerical data can be found in S1 Data. The dataset, including the raw images, the tracking files, and all the CellTracksColab results (including numerical data), are also available on Zenodo (11286110).

MYO10 were allowed to migrate either beneath a collagen gel or within standard media (data describing the migration behavior of these cells in standard media was previously reported here [19]). Automated cell tracking was achieved using StarDist and TrackMate algorithms [1]. This larger dataset encompasses 117 fields of view (videos) spread across 4 conditions and 3 biological repeats. CellTracksColab compiled this dataset in around 6 minutes using Colab, storing 49,268 tracks and 3,262,747 tracked objects.

As with the T cell dataset, we first performed quality control steps after computing additional track metrics. In the case of the breast cancer cell dataset, our quality control revealed some challenges: First, the third biological repeat (R3) did not cluster with the others (S3A Fig). Additionally, the dataset was unbalanced, with the third repeat contributing disproportionately more tracks (Fig 3B). Together, this analysis could indicate an issue with this R3 and signal that the experiments might need to be repeated a fourth time. In addition, given this imbalance, we deemed it imperative to resample the dataset to ensure that R3's data do not unduly influence the overall conclusions. To ensure the robustness of the resampling, Cell-TracksColab allows for performing a statistical comparison between the original and resampled data per condition and track metric. The outcomes of this comparison are succinctly visualized in a heatmap, providing a clear and accessible way to assess the effects of resampling on the dataset's overall distribution (S3B Fig). Post-resampling, the dataset contains 1,337 tracks for each condition and repeat (S3C Fig).

In our analysis of the resampled dataset using CellTracksColab, we focused on key track metrics such as mean speed and directionality to elucidate patterns of collective migration. We find that shMYO10 cells migrate slower than control cells, a pattern that persists with cells migrating beneath the collagen gel (Figs 3C and S4A). Intriguingly, a closer examination of individual tracks revealed that cells in the R3 exhibited a faster movement, yet this did not alter the overall migration differences observed in the dataset (Fig 3C). Without a collagen gel, we observed no significant differences in the directionality of migration between the conditions (Figs 3C and S4B). However, MYO10-silenced cells displayed increased directionality under the collagen gel compared to control cells, which was an unexpected finding (Fig 3C).

Further analysis utilizing 2D UMAP projections of the whole dataset revealed challenges in cluster generation likely due to the similarity in track characteristics, a common occurrence in collective migration (Fig 3D). Nevertheless, by employing the Canberra distance, distinct clusters were successfully delineated (S4C Fig). These clusters provided a clear "fingerprint" for each condition, with cluster 2 highlighting key differences between conditions (Fig 3E). Cluster 2 is characterized by tracks of low speed but high directionality (S4D Fig). Within cluster 2, MYO10-silenced cells showed increased directionality compared to CTRL cells in the presence of a collagen gel (S4E Fig). Additionally, CTRL cells in this cluster moved faster than their MYO10-silenced counterparts (S4F Fig).

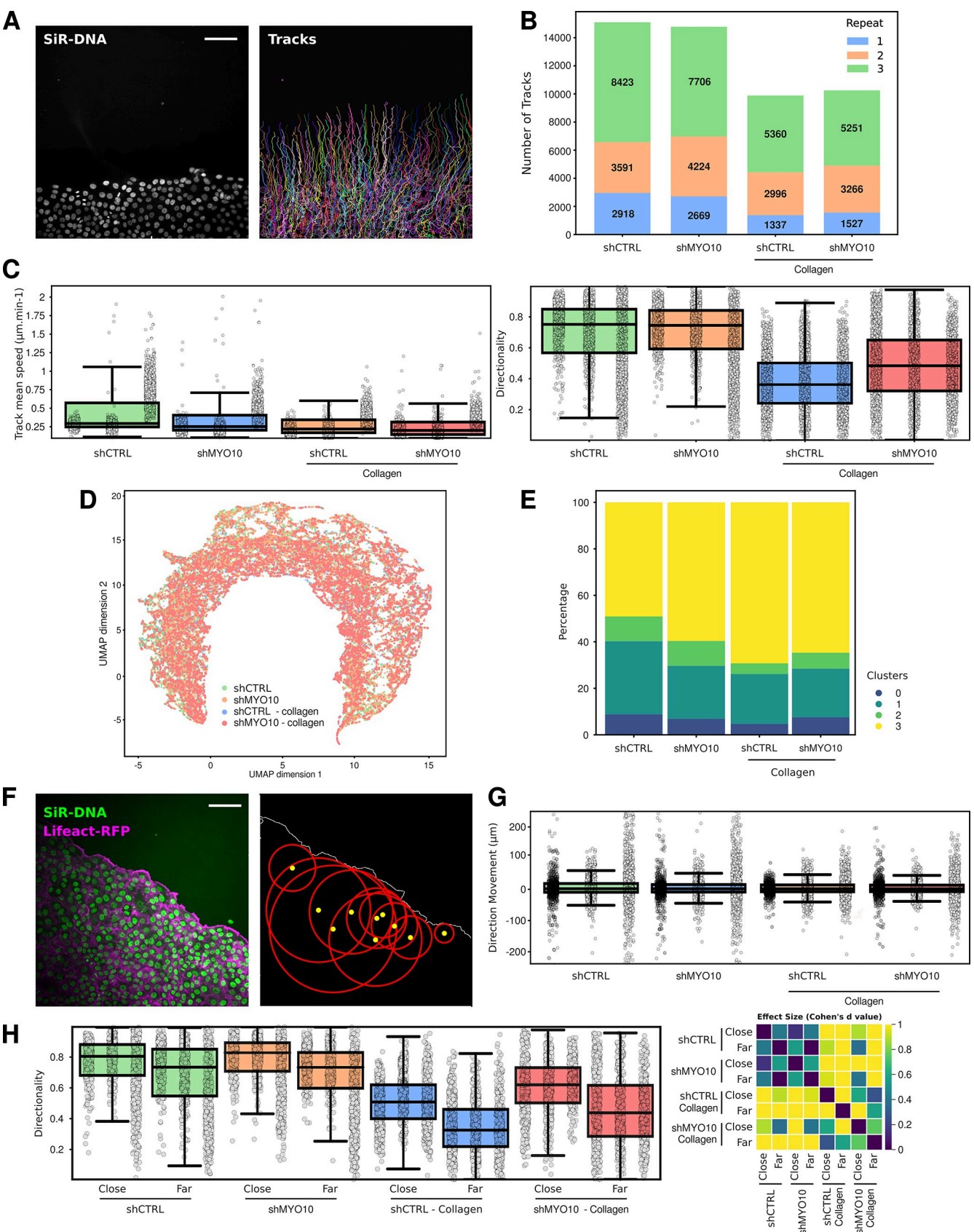

**Fig 3. Exploring cancer cell migration using CellTracksColab.** (**A**) MCF10DCIS.com lifeact-RFP cells, labeled with SiR-DNA, were recorded live using a spinning disk confocal microscope and tracked using StarDist and TrackMate. Detected nuclei and tracks (colors indicate track ID) are displayed. Scale bar: 100 µm. (**B**) This panel presents a stacked histogram showcasing the number of tracks for each biological repeat under different conditions. Each biological repeat is color-coded, and each histogram segment's specific number of tracks is annotated. (**C**) The "track mean speed" and "directionality" metrics for each condition (resampled dataset) are summarized in a Tukey boxplot format. The *p*-value and Cohen's *d* value heatmaps are shown in S4A and S4B Fig. (**D**) 2D UMAP projection of the entire dataset, using all available track metrics for dimensionality reduction. Data points are color-coded based on the conditions. (**E**) A fingerprint plot showcasing the distribution percentage of each cluster across different conditions (clustering is shown in S4C Fig). (**F**) Distance measurement of 10 selected tracks to the monolayer leading edge. The image on the left is the raw microscopy image. The image on the right was generated by CellTracksColab to visually validate that the distance measurements are correct. The yellow dots indicate the randomly selected tracks, and the red circles indicate the measured distance. The leading edge is displayed in white. Scale bar: 100 µm. (**G**) The "Direction Movement" metric for each condition (whole dataset) is summarized in a Tukey boxplot format. This metric is calculated as EndDistance—StartDistance (the distances of the track from the leading edge at the end and the start of the tracking period, respectively). A positive value indicates moving away from the leading edge over time, and a negative value suggests moving closer. The *p*-value and Cohen's *d* value heatmaps are shown in S4G Fig. (**H**) After separating the tracks based on their maximal distance to the leading edge (close, distance < 75 µm; far, distance > 75 µm), the track "directionality" metric for each condition (whole dataset) was summarized in a Tukey boxplot. The mirrored heatmap displaying the Cohen's *d* value between each condition is shown on the right. For all box plots, the vertical whiskers extend to data points within 1.5× the interquartile range, and the values for each track are shown as dots. Each biological replicate is displayed next to each other from R1 to R3 (left to right). Plot axes are limited to 10× the interquartile range. (**C**, **G**, and **H**) Underlying numerical data can be found in S1 Data. The dataset, including the raw images, the tracking files, and all the CellTracksColab results (including numerical data), are also available on Zenodo (11282716).

In the study of collective migration, a critical aspect to consider is the distinct behavior of leading cells compared to those positioned further from the leading edge [20]. To address this, we used the CellTracksColab spatial analysis module to measure each track's distance to the leading edge over time (Fig 3F). Interestingly, overall, we observed no significant differences among the conditions in the cells' capability to target the leading edge (Figs 3G and S4E). Yet, the "Direction Movement" metric revealed a complex scenario: While, on average, cell distance to the leading edge does not change between the beginning and the end of the track, the data distribution was broad. Many tracks were found closer to the leading edge by the end of the tracking period, while others were found to be further away (Fig 3G).

Delving deeper, CellTracksColab allowed us to segregate the tracks based on their maximum distance to the leading edge, distinguishing between tracks proximal to and distant from the monolayer edge. This stratified analysis unveiled that, across all conditions, cells closer to the leading edge exhibited more directional movement compared to those further away (Fig 3H). Notably, in the presence of a collagen gel, silencing MYO10 resulted in more directional movement, irrespective of the cells' proximity to the leading edge (Fig 3H). This example highlights how CellTracksColab can help extract spatial insights from tracking data. Furthermore, the spatial metrics derived from these analyses can enrich dimensionality reduction analyses, potentially helping unveil additional nuanced behaviors in tracking data.

## Studying filopodia dynamics using CellTracksColab

In our final example, we aimed to showcase the versatility of CellTracksColab by exploring a filopodia dynamics dataset [21], diverging from our previous focus on cell migration. This study involved U2OS cells expressing different MYO10 constructs, a protein-inducing filopodia formation that also accumulates at their tips [22] (Fig 4A). We tracked MYO10 puncta in live cells to investigate the dynamics of filopodia induced by 3 MYO10 variants: the wild type (MYO10$^{WT}$), a mutant lacking the MyTH4/FERM domain (MYO10$^{\Delta FERM}$), and a chimera (MYO10$^{TH}$), where MYO10's FERM domain is replaced by that from TLN1 [21]. The dataset encompasses 3 experimental conditions, each with 3 biological replicates and a total of 112 videos. Utilizing CellTracksColab in Colab, we efficiently compiled this dataset, which included 91,825 tracks and nearly 1.5 million tracked objects, in around 4 minutes.

We started the analysis by filtering out tracks lasting for less than 25 seconds, resulting in a refined dataset comprising 57,487 tracks and 1,377,019 objects. Utilizing UMAP coupled with clustering analysis (Figs 4B and S5A), we identified several distinct clusters, providing a

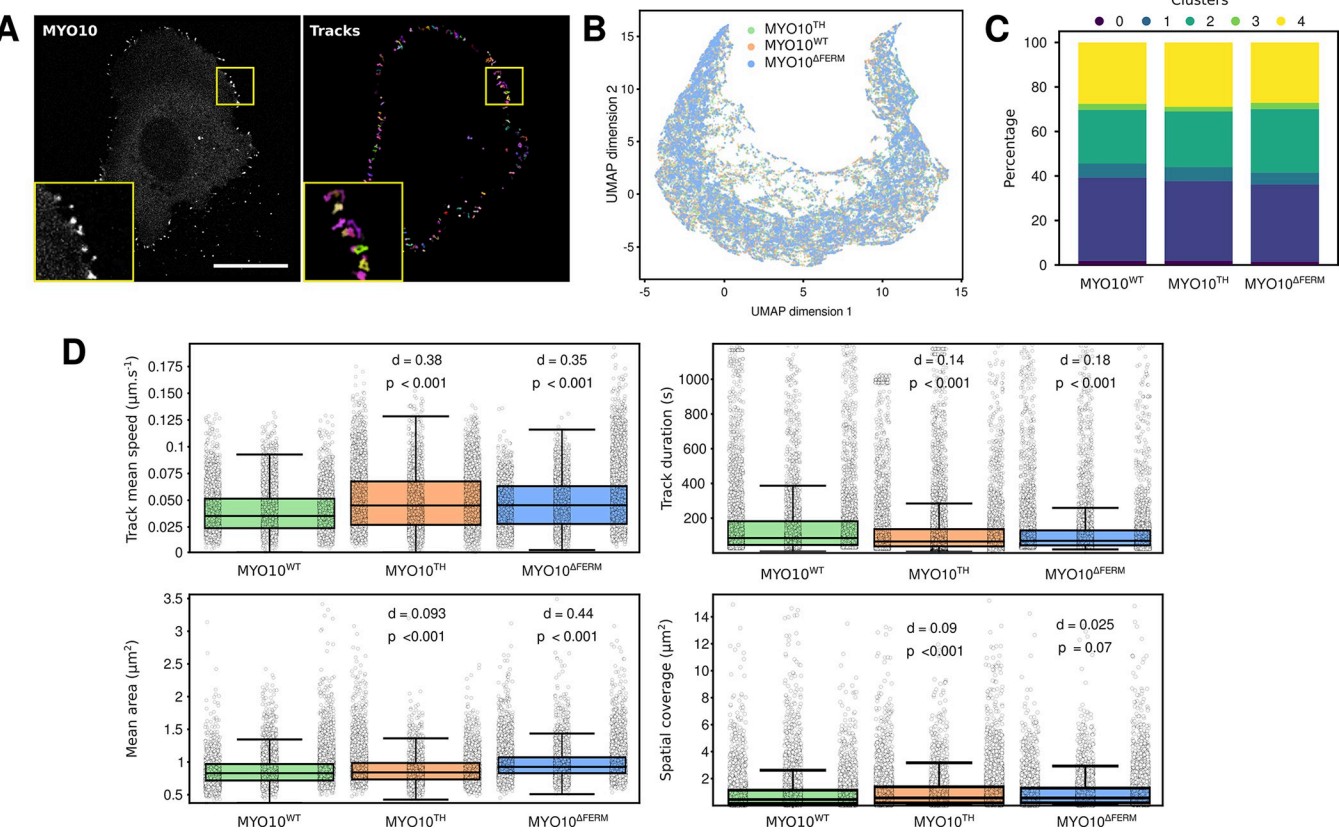

**Fig 4. Exploring filopodia dynamics using CellTracksColab.** (**A**) A U2OS cell expressing MYO10WT-GFP was imaged live using an Airyscan confocal microscope. MYO10 puncta were then tracked using StarDist and TrackMate. Detected MYO10 puncta and tracks (colors indicate track ID) are displayed. Scale bar: 25 μm. (**B**) 2D UMAP projection of the entire dataset, using all available track metrics for dimensionality reduction. Data points are color-coded based on the conditions. (**C**) A fingerprint plot showcasing the distribution percentage of each cluster across different conditions (clustering is shown in S5A Fig). (**D**) The "track mean speed," the "track duration," the spot "mean area," and the track "spatial coverage" for each condition are summarized in Tukey boxplots. The effect size (d, Cohen's *d* value) and the statistical significance (p, *p*-values from randomization tests) between the MYO10$^{WT}$ and the indicated conditions are displayed. For all box plots, the vertical whiskers extend to data points within 1.5× the interquartile range, and the values for each track are shown as dots. Each biological replicate is displayed next to each other from R1 to R3 (left to right). Plot axes are limited to 10× the interquartile range. Underlying numerical data can be found in S1 Data. The dataset, including the raw images, the tracking files, and all the CellTracksColab results (including numerical data), are also available on Zenodo (11285514).

window into the intricate behaviors of filopodia (S5B Fig). However, the fingerprint plot revealed a similar distribution of tracks across clusters within each experimental condition (Fig 4C). Given our focus on differences between the MYO10 constructs, we did not extensively investigate individual clusters.

Quality control assessments highlighted that the biological repeats were not clustering cohesively and revealed an imbalance in the dataset across both repeats and conditions (S5C and S5D Fig). Consequently, we resampled the data to ensure a more balanced representation before proceeding with the analysis of track metrics.

Post-resampling, we observed notable distinctions: MYO10$^{WT}$ filopodia exhibited greater stability, evidenced by longer lifetimes (track duration) and slower speeds, compared to MYO10$^{\Delta FERM}$ and MYO10$^{TH}$ filopodia (Fig 4D). Interestingly, MYO10$^{\Delta FERM}$ filopodia had a larger area than MYO10$^{WT}$, and while MYO10$^{TH}$ filopodia also showed a statistically significant difference in area from MYO10$^{WT}$ (*p*-value < 0.001), the low Cohen's *d* value suggests a negligible practical difference between these conditions (Fig 4D). This observation highlights

the importance of using both Cohen's $d$ and $p$-values when comparing conditions, as it provides a more nuanced understanding of the data.

## Discussion

Here, we introduced the CellTracksColab platform, a tool designed for the life sciences community that offers a user-friendly solution for tracking data analysis. CellTracksColab integrates several functionalities for tracking data analysis, including track visualization, population analysis, statistical assessments, and dimensionality reduction.

The easiest way to start using CellTracksColab is via the Google Colaboratory framework, which significantly simplifies access and overcomes common barriers such as complex software installations. Its intuitive graphical user interface effectively bridges the gap between sophisticated computational methods and researchers with limited programming skills, making it more inclusive. However, it is essential to acknowledge the limitations of the Google Colaboratory environment. One of the primary constraints is the limited runtime, as sessions in Colab are typically capped, which can interrupt longer analytical processes. Additionally, the computing power (especially the runtime RAM) and speed offered by the free version of Colab may not suffice for massive datasets. There are also concerns regarding data privacy. To mitigate these issues, CellTracksColab can operate locally via Jupyter notebooks. Running the platform locally enables users to utilize their computational resources, providing extended runtime, increased processing power, and better control over data privacy. However, in a standard Jupyter Notebook environment, the code is exposed by default, which might make the interface seem less streamlined compared to the encapsulated Colab version. For those who prefer the Colab interface but want to use their local machine's resources, connecting Google Colab to a local Python environment is a viable option. This hybrid approach leverages the familiar Colab interface while utilizing local computational power. We believe that these 3 modalities—Google Colaboratory, local Jupyter notebooks, and a hybrid local Colab connection—provide comprehensive options to accommodate the preferences and needs of most users.

Existing image repositories, such as the BioImage Archive [23] and the Image Data Resource [24], demonstrate the feasibility and value of sharing microscopy data. Analyzed tracking datasets are also very valuable; they hold significant potential for reanalysis and meta-analysis and offer a more manageable alternative to storing raw video footage [25]. Moreover, they can serve the purpose of new machine learning tracking algorithm development and benchmarking. Yet, their widespread sharing remains limited, and publicly available analyzed datasets are relatively scarce. This scarcity is partly due to the need for a standardized format for sharing tracking results. CellTracksColab partially addresses this challenge by adopting a unified format for storing tracking data and simplifying the sharing of tracking datasets. Additionally, the platform includes a streamlined notebook designed for easy loading, viewing, and replotting previously analyzed datasets. This feature enhances data analysis' transparency and promotes reproducibility, aligning with the FAIR (Findable, Accessible, Interoperable, and Reusable) principles in scientific research.

Despite its robust capabilities, CellTracksColab has certain limitations we aim to address in future updates. Currently, the platform has limited support for analyzing tracks in 3D space. While the platform can handle 3D + time datasets for metric computation, quality control, and dimensionality reduction tasks, it falls short in certain areas. Specifically, some of its analysis modules, including track visualization and spatial analysis, are optimized only for 2D + time datasets. In addition, CellTracksColab is not currently adapted for lineage tracing

studies. Researchers focusing on lineage tracing may explore alternative tools specifically developed for such analyses [1,3,26].

We also plan to enhance CellTracksColab by introducing additional analytical features. This includes the capability to examine time-series data, such as analyzing variations in fluorescent reporters within tracked objects over time, which could provide deeper insights into dynamic biological processes [27,28]. Furthermore, we intend to extend our data loader for other popular tracking tools, broadening the platform's compatibility and ease of integration with existing workflows. In conclusion, we believe that CellTracksColab represents a significant step forward in tracking data analysis in life sciences. Its user-friendly design and robust analytical capabilities allow researchers to explore and understand the complexities of biological motion and behavior. As we continue to develop and enhance CellTracksColab, we anticipate it becoming a useful tool in the life sciences toolkit, aiding in discovering and understanding new biological insights.

## Materials and methods

### Implementation

CellTracksColab is implemented as a series of interactive Jupyter notebooks. The platform utilizes Python as its primary programming language, leveraging various libraries such as Pandas for data manipulation, Matplotlib, Seaborn for data visualization, and UMAP and HDBSCAN for dimensionality reduction and clustering analyses. The notebook's architecture facilitates ease of use while providing robust data analysis capabilities. The notebooks are structured to guide users through each step of the analysis process, from data import and preprocessing to advanced statistical and spatial analyses. Each notebook contains detailed instructions and documentation to assist users in customizing the analysis to their specific datasets.

### Data and software availability

Multiple test datasets are available on Zenodo. They include the 2 test datasets that can be directly downloaded from within the CellTracksColab notebooks (8413510, 8420011) [29,30]. In addition, the 3 datasets showcased in this study, their tracking files, and the CellTracksColab results are also available on Zenodo (11282716, 11286110, 11285514) [31–33].

The code for CellTracksColab is publicly available under the MIT license, encouraging broad utilization and adaptation. CellTracksColab's GitHub repository serves as a dynamic platform for tracking the evolution of the code across various versions. Users are encouraged to report issues and suggest features directly through the GitHub interface. A stable version of the code and associated documentation is also archived on Zenodo (11384844) [34].

### The T cell dataset

The T cell dataset used is available on Zenodo (11286110) [32] and has been detailed in previous publications [1,17,18]. In summary, Lab-Tek 8 chamber slides (Thermo Fisher) were prepared by overnight coating with either 2 μg/mL ICAM-1 or VCAM-1 at a temperature of 4°C. Subsequently, activated primary mouse CD4+ T cells were cleansed and suspended in L-15 media, enriched with 2 mg/mL D-glucose. These T cells were then placed into the chamber slides and incubated for 20 minutes. Post-incubation, a gentle wash was performed to eliminate all unattached cells. The imaging process was conducted using a 10× phase contrast objective at 37°C, utilizing a Zeiss Axiovert 200M microscope equipped with an automated X-Y stage and a Roper EMCCD camera. Time-lapse imaging was executed at intervals of 1 minute over 10 minutes, employing SlideBook 6 software from Intelligent Imaging Innovations.

Cells were automatically tracked using StarDist, directly called within TrackMate [1,15,35]. The StarDist model was trained using ZeroCostDL4Mic [11] and is publicly available on Zenodo (4034929) [36]. This model generated excellent segmentation results on our test data-set (F1 score > 0.99). In TrackMate, the StarDist detector custom model (score threshold = 0.41 and overlap threshold = 0.5) and the Simple LAP tracker (linking max distance = 30 μm; gap closing max distance = 15 μm, gap closing max frame gap = 2 frames) were used.

In CellTracksColab, we conducted a dimensionality reduction analysis employing UMAP. The UMAP settings were as follows: number of neighbors (*n_neighbors*) set to 10, minimum distance (*min_dist*) to 0, and number of dimensions (*n_dimension*) to 2. This analysis utilized an array of track metrics, including the following:

```
NUMBER_SPOTS, NUMBER_GAPS, NUMBER_SPLITS, NUMBER_MERGES, NUMBER_COM-
PLEX, LONGEST_GAP, TRACK_DISPLACEMENT, TRACK_MEAN_QUALITY, MAX_DIS-
TANCE_TRAVELED, CONFINEMENT_RATIO, MEAN_STRAIGHT_LINE_SPEED,
LINEARITY_OF_FORWARD_PROGRESSION, MEAN_DIRECTIONAL_CHANGE_RATE, Track
Duration, Mean Speed, Median Speed, Max Speed, Min Speed, Speed Stan-
dard Deviation, Total Distance Traveled, Directionality, Tortuosity,
MEAN_CIRCULARITY, MEAN_SOLIDITY, MEAN_SHAPE_INDEX, MEDIAN_CIRCULAR-
ITY, MEDIAN_SOLIDITY, MEDIAN_SHAPE_INDEX, STD_CIRCULARITY, STD_SOLID-
ITY, STD_SHAPE_INDEX, MIN_CIRCULARITY, MIN_SOLIDITY, MIN_SHAPE_INDEX,
MAX_CIRCULARITY, MAX_SOLIDITY, MAX_SHAPE_INDEX
```

Subsequently, clustering analysis was performed using HDBSCAN. The parameters included *clustering_data_source* set to UMAP, *min_samples* at 20, *min_cluster_size* at 200, and the metric employed was Euclidean.

## The breast cancer cell dataset

The tracked breast cancer cell dataset is available on Zenodo (11282716) [31]. In this experi-ment, approximately 50,000 shCTRL or shMYO10 lifeact-RFP DCIS.COM cells [19] were seeded into 1 well of an ibidi culture-insert 2 well preplaced in a μ-Slide 8 well. The cells were cultured for 24 hours, after which the culture insert was removed to create a wound-healing assay setup. When appropriate, a fibrillar collagen gel (PureCol EZ Gel) was applied over the cells and allowed to polymerize for 30 minutes at 37˚C. Standard culture media was added to all wells, and the cells were left to migrate/invade for 2 days [37]. Before live cell imaging, the cells were treated with 0.5 μM SiR-DNA (SiR-Hoechst, Tetu-bio) for 2 hours. Imaging was performed over 14 hours using a Marianas spinning-disk confocal microscope system. This system included a Yokogawa CSU-W1 scanning unit mounted on an inverted Zeiss Axio Observer Z1 microscope (Intelligent Imaging Innovations). Imaging was conducted using a 20× (NA 0.8) air Plan Apochromat objective (Zeiss), and images were captured at 10-minute intervals.

Cell tracking was conducted using Fiji [15] and TrackMate [1]. The StarDist detector was employed to detect nuclei using the StarDist versatile model [35]. Tracks were created using the Kalman tracker (a maximum frame gap of 1, a Kalman search radius of 20 μm, and a link-ing maximum distance of 15 μm). Post-tracking, tracks were filtered so that each track had to contain more than 6 spots, ensuring a significant amount of data per track, and the total dis-tance traveled by cells had to be greater than 89 μm.

In CellTracksColab, we conducted a dimensionality reduction analysis employing UMAP. The UMAP settings were as follows: number of neighbors (n_neighbors) set to 10, minimum distance (min_dist) to 0, and number of dimensions (n_dimension) to 2. This analysis utilized an array of track metrics, including the following:

```
NUMBER_SPOTS, NUMBER_GAPS, NUMBER_SPLITS, NUMBER_MERGES, NUMBER_COM-
PLEX, LONGEST_GAP, TRACK_DISPLACEMENT, TRACK_MEAN_QUALITY, MAX_DISTAN-
CE_TRAVELED, CONFINEMENT_RATIO, MEAN_STRAIGHT_LINE_SPEED,
LINEARITY_OF_FORWARD_PROGRESSION, MEAN_DIRECTIONAL_CHANGE_RATE, Track
Duration, Mean Speed, Median Speed, Max Speed, Min Speed, Speed Stan-
dard Deviation, Total Distance Traveled, Directionality, Tortuosity,
Total Turning Angle, Spatial Coverage, MEAN_MEAN_INTENSITY_CH1, MEAN_-
MEDIAN_INTENSITY_CH1, MEAN_MIN_INTENSITY_CH1, MEAN_MAX_INTENSITY_CH1,
MEAN_TOTAL_INTENSITY_CH1, MEAN_STD_INTENSITY_CH1, MEAN_CONTRAST_CH1,
MEAN_SNR_CH1, MEAN_ELLIPSE_X0, MEAN_ELLIPSE_Y0, MEAN_ELLIPSE_MAJOR,
MEAN_ELLIPSE_MINOR, MEAN_ELLIPSE_THETA, MEAN_ELLIPSE_ASPECTRATIO,
MEAN_AREA, MEAN_PERIMETER, MEAN_CIRCULARITY, MEAN_SOLIDITY, MEAN_SHA-
PE_INDEX, MEDIAN_MEAN_INTENSITY_CH1, MEDIAN_MEDIAN_INTENSITY_CH1, MED-
IAN_MIN_INTENSITY_CH1, MEDIAN_MAX_INTENSITY_CH1,
MEDIAN_TOTAL_INTENSITY_CH1, MEDIAN_STD_INTENSITY_CH1, MEDIAN_CON-
TRAST_CH1, MEDIAN_SNR_CH1, MEDIAN_ELLIPSE_X0, MEDIAN_ELLIPSE_Y0, MED-
IAN_ELLIPSE_MAJOR, MEDIAN_ELLIPSE_MINOR, MEDIAN_ELLIPSE_THETA,
MEDIAN_ELLIPSE_ASPECTRATIO, MEDIAN_AREA, MEDIAN_PERIMETER, MEDIAN_CIR-
CULARITY, MEDIAN_SOLIDITY, MEDIAN_SHAPE_INDEX, STD_MEAN_INTENSITY_CH1,
STD_MEDIAN_INTENSITY_CH1, STD_MIN_INTENSITY_CH1, STD_MAX_INTENSI-
TY_CH1, STD_TOTAL_INTENSITY_CH1, STD_STD_INTENSITY_CH1, STD_CON-
TRAST_CH1, STD_SNR_CH1, STD_ELLIPSE_X0, STD_ELLIPSE_Y0,
STD_ELLIPSE_MAJOR, STD_ELLIPSE_MINOR, STD_ELLIPSE_THETA, STD_ELLIP-
SE_ASPECTRATIO, STD_AREA, STD_PERIMETER, STD_CIRCULARITY, STD_SOLID-
ITY, STD_SHAPE_INDEX, MIN_MEAN_INTENSITY_CH1,
MIN_MEDIAN_INTENSITY_CH1, MIN_MIN_INTENSITY_CH1, MIN_MAX_INTENSI-
TY_CH1, MIN_TOTAL_INTENSITY_CH1, MIN_STD_INTENSITY_CH1, MIN_CON-
TRAST_CH1, MIN_SNR_CH1, MIN_ELLIPSE_X0, MIN_ELLIPSE_Y0,
MIN_ELLIPSE_MAJOR, MIN_ELLIPSE_MINOR, MIN_ELLIPSE_THETA, MIN_ELLIP-
SE_ASPECTRATIO, MIN_AREA, MIN_PERIMETER, MIN_CIRCULARITY, MIN_SOLID-
ITY, MIN_SHAPE_INDEX, MAX_MEAN_INTENSITY_CH1,
MAX_MEDIAN_INTENSITY_CH1, MAX_MIN_INTENSITY_CH1, MAX_MAX_INTENSI-
TY_CH1, MAX_TOTAL_INTENSITY_CH1, MAX_STD_INTENSITY_CH1, MAX_CON-
TRAST_CH1, MAX_SNR_CH1, MAX_ELLIPSE_X0, MAX_ELLIPSE_Y0,
MAX_ELLIPSE_MAJOR, MAX_ELLIPSE_MINOR, MAX_ELLIPSE_THETA, MAX_ELLIP-
SE_ASPECTRATIO, MAX_AREA, MAX_PERIMETER, MAX_CIRCULARITY, MAX_SOLID-
ITY, MAX_SHAPE_INDEX, MaxDistance_edge, MinDistance_edge,
StartDistance_edge, EndDistance_edge, MedianDistance_edge, StdDevDis-
tance_edge, DirectionMovement_edge, AvgRateChange_edge, Percentage-
Change_edge, TrendSlope_edge
```

Subsequently, clustering analysis was performed using HDBSCAN. The parameters included clustering_data_source set to UMAP, min_samples at 20, min_cluster_size at 200, and the metric employed was Canberra.

## The filopodia dataset

The tracked filopodia dataset is available on Zenodo (11285514) [33] and was previously described [21]. U2OS cells expressing MYO10-GFP, a MYO10 MyTH/FERM deletion construct (EGFP-MYO10$^{\Delta FERM}$), or an MYO10/TLN1 chimera construct (EGFP-MYO10$^{TH}$) were plated for at least 2 hours on fibronectin before the start of live imaging. Images were taken every 5 seconds at 37°C on an Airyscan microscope, using a 40× objective.

MYO10 puncta were tracked using TrackMate using the custom StarDist detector and the simple LAP tracker (Linking max distance = 1 μm, Gap-closing max distance = 2 μm, Gap-closing max frame gap = 2 μm). The StarDist 2D model used was previously described [38].

Briefly, this model was trained for 200 epochs on 11 paired image patches [image dimensions: (512, 512), patch size: (512,512)] with a batch size of 2 and a mean absolute error (MAE) loss function, using the StarDist 2D ZeroCostDL4Mic notebook [11].

In CellTracksColab, we conducted a dimensionality reduction analysis employing UMAP. The UMAP settings were as follows: number of neighbors (n_neighbors) set to 10, minimum distance (min_dist) to 0, and number of dimensions (n_dimension) to 2. This analysis utilized an array of track metrics, including the following:

```
NUMBER_SPOTS, NUMBER_GAPS, NUMBER_SPLITS, NUMBER_MERGES, NUMBER_COM-
PLEX, LONGEST_GAP, TRACK_DISPLACEMENT, TRACK_MEAN_QUALITY, MAX_DISTAN-
CE_TRAVELED, CONFINEMENT_RATIO, MEAN_STRAIGHT_LINE_SPEED,
LINEARITY_OF_FORWARD_PROGRESSION, MEAN_DIRECTIONAL_CHANGE_RATE, Track
Duration, Mean Speed, Median Speed, Max Speed, Min Speed, Speed Stan-
dard Deviation, Total Distance Traveled, Directionality, Tortuosity,
Total Turning Angle, Spatial Coverage, MEAN_MEAN_INTENSITY_CH1, MEAN_-
MEDIAN_INTENSITY_CH1, MEAN_MIN_INTENSITY_CH1, MEAN_MAX_INTENSITY_CH1,
MEAN_TOTAL_INTENSITY_CH1, MEAN_STD_INTENSITY_CH1, MEAN_CONTRAST_CH1,
MEAN_SNR_CH1, MEAN_ELLIPSE_X0, MEAN_ELLIPSE_Y0, MEAN_ELLIPSE_MAJOR,
MEAN_ELLIPSE_MINOR, MEAN_ELLIPSE_THETA, MEAN_ELLIPSE_ASPECTRATIO,
MEAN_AREA, MEAN_PERIMETER, MEAN_CIRCULARITY, MEAN_SOLIDITY, MEAN_SHA-
PE_INDEX, MEDIAN_MEAN_INTENSITY_CH1, MEDIAN_MEDIAN_INTENSITY_CH1, MED-
IAN_MIN_INTENSITY_CH1, MEDIAN_MAX_INTENSITY_CH1,
MEDIAN_TOTAL_INTENSITY_CH1, MEDIAN_STD_INTENSITY_CH1, MEDIAN_CON-
TRAST_CH1, MEDIAN_SNR_CH1, MEDIAN_ELLIPSE_X0, MEDIAN_ELLIPSE_Y0, MED-
IAN_ELLIPSE_MAJOR, MEDIAN_ELLIPSE_MINOR, MEDIAN_ELLIPSE_THETA,
MEDIAN_ELLIPSE_ASPECTRATIO, MEDIAN_AREA, MEDIAN_PERIMETER, MEDIAN_CIR-
CULARITY, MEDIAN_SOLIDITY, MEDIAN_SHAPE_INDEX, STD_MEAN_INTENSITY_CH1,
STD_MEDIAN_INTENSITY_CH1, STD_MIN_INTENSITY_CH1, STD_MAX_INTENSI-
TY_CH1, STD_TOTAL_INTENSITY_CH1, STD_STD_INTENSITY_CH1, STD_CON-
TRAST_CH1, STD_SNR_CH1, STD_ELLIPSE_X0, STD_ELLIPSE_Y0,
STD_ELLIPSE_MAJOR, STD_ELLIPSE_MINOR, STD_ELLIPSE_THETA, STD_ELLIP-
SE_ASPECTRATIO, STD_AREA, STD_PERIMETER, STD_CIRCULARITY, STD_SOLID-
ITY, STD_SHAPE_INDEX, MIN_MEAN_INTENSITY_CH1,
MIN_MEDIAN_INTENSITY_CH1, MIN_MIN_INTENSITY_CH1, MIN_MAX_INTENSI-
TY_CH1, MIN_TOTAL_INTENSITY_CH1, MIN_STD_INTENSITY_CH1, MIN_CON-
TRAST_CH1, MIN_SNR_CH1, MIN_ELLIPSE_X0, MIN_ELLIPSE_Y0,
MIN_ELLIPSE_MAJOR, MIN_ELLIPSE_MINOR, MIN_ELLIPSE_THETA, MIN_ELLIP-
SE_ASPECTRATIO, MIN_AREA, MIN_PERIMETER, MIN_CIRCULARITY, MIN_SOLID-
ITY, MIN_SHAPE_INDEX, MAX_MEAN_INTENSITY_CH1,
MAX_MEDIAN_INTENSITY_CH1, MAX_MIN_INTENSITY_CH1, MAX_MAX_INTENSI-
TY_CH1, MAX_TOTAL_INTENSITY_CH1, MAX_STD_INTENSITY_CH1, MAX_CON-
TRAST_CH1, MAX_SNR_CH1, MAX_ELLIPSE_X0, MAX_ELLIPSE_Y0,
MAX_ELLIPSE_MAJOR, MAX_ELLIPSE_MINOR, MAX_ELLIPSE_THETA, MAX_ELLIP-
SE_ASPECTRATIO, MAX_AREA, MAX_PERIMETER, MAX_CIRCULARITY, MAX_SOLID-
ITY, MAX_SHAPE_INDEX.
```

Subsequently, clustering analysis was performed using HDBSCAN. The parameters included clustering_data_source set to UMAP, min_samples at 20, min_cluster_size at 600, and the metric employed was Canberra.

### Manuscript preparation

GPT-4 (OpenAI) and Grammarly (Grammarly) were used as writing aids while preparing this manuscript. All text sections generated were further edited and validated by the author. No references were provided by GPT-4. All figure panels (except Figs 2A, 3A and 4A) and all

statistical analyses were generated directly within CellTracksColab and edited in Inkscape (Inkscape Project).

## Supporting information

**S1 Fig. Track visualization and filtering.** Example of track display before (**A**) and after (**B**) smoothing and filtering. The tracks originate from a video of migrating breast cancer cells tracked from their nuclei using TrackMate.
(TIF)

**S2 Fig. Evaluating the experimental variability in the T cell dataset with CellTracksColab.** (**A**) This panel presents a stacked histogram showcasing the number of tracks for each biological repeat under different conditions, aiding in evaluating the dataset's balance. Each biological repeat is color-coded, and each histogram segment's specific number of tracks is annotated. (**B**, **C**) Hierarchical clustering: These dendrograms reveal the hierarchical clustering within the dataset by utilizing the cosine similarity metric and a complete linkage method. (**B**) FOV-based clustering analysis: This dendrogram illustrates the clustering across the 10 available fields of view (FOVs). (**C**) Condition and repeat-based clustering: This dendrogram delves deeper by segregating the dataset based on conditions and biological repeats. The dataset, including the raw images, the tracking files, and all the CellTracksColab results (including numerical data), are also available on Zenodo (11282716).
(TIF)

**S3 Fig. Evaluating the experimental variability in the breast cancer cell dataset with Cell-TracksColab.** (**A**) This dendrogram utilizes the cosine similarity metric and a complete linkage method to assess the similarity in the dataset between conditions and biological repeats. (**B**) $p$-Value heatmap comparing the differences between the data distribution before and after resampling for each condition and repeats (selected number of track metrics). (**C**) This panel presents a stacked histogram showcasing the number of tracks for each biological repeat under different conditions, aiding in evaluating the dataset's balance. Each biological repeat is color-coded, and each histogram segment's specific number of tracks is annotated. The dataset, including the raw images, the tracking files, and all the CellTracksColab results (including numerical data), are also available on Zenodo (11282716).
(TIF)

**S4 Fig. Exploring the breast cancer migration dataset using CellTracksColab.** (**A**, **B**) $p$-Value and Cohen's $d$ value mirrored heatmaps for the "track mean speed" (**A**) and track "directionality" (**B**) metrics (see Fig 3C). (**C**) 2D UMAP projection of the entire breast cancer migration dataset, using all available track metrics for dimensionality reduction. Resultant clusters from the HDBSCAN analysis on the 2D UMAP projection. The Canberra method served as the metric for clustering. Each identified cluster is color-coded. (**D**) The "track mean speed" and track "directionality" for each cluster are summarized in a Tukey boxplot format. (**E**, **F**) The track "directionality" (**E**) and "track mean speed" (**F**) metrics for each condition for cluster 2 are summarized in a Tukey boxplot format. For all box plots, the vertical whiskers extend to data points within 1.5× the interquartile range, and the values for each track are shown as dots where each biological replicate is displayed next to each other from R1 to R3 (left to right). $p$-Value and Cohen's $d$ value mirrored heatmaps are displayed on the right. (**G**) $p$-Value and Cohen's $d$ value mirrored heatmaps for the "Direction Movement" metric (see Fig 3G). (**D**, **E,** and **F**) Underlying numerical data can be found in S1 Data. The dataset, including the raw images, the tracking files, and all the CellTracksColab results (including

numerical data), are also available on Zenodo (11282716).
(TIF)

**S5 Fig. Exploring filopodia dynamics using CellTracksColab.** (**A**) 2D UMAP projection of the entire filopodia dataset, using all available track metrics for dimensionality reduction. Resultant clusters from the HDBSCAN analysis on the 2D UMAP projection. The Canberra method served as the metric for clustering. Each identified cluster is color-coded. (**B**) Heatmap representation, normalized using Z-scores, displaying variations in selected track metrics among the clusters. Full heatmaps are available in the Zenodo archive of this dataset. (**C**) This dendrogram utilizes the cosine similarity metric and a complete linkage method to assess the similarity in the filopodia dataset between conditions and biological repeats. (**D**) This panel presents a stacked histogram showcasing the number of tracks for each biological repeat under different conditions. Each biological repeat is color-coded, and each histogram segment's specific number of tracks is annotated. The dataset, including the raw images, the tracking files, and all the CellTracksColab results (including numerical data), are also available on Zenodo (11285514).
(TIF)

**S1 Data. Underlying numerical data for all the box plots shown in the paper.** The data are stored in the tiddy format. All datasets showcased in this study, their tracking files, and the CellTracksColab results are also available on Zenodo (11282716, 11286110, 11285514).
(XLSX)

## Acknowledgments

We thank Hellyeh Hamidi for providing feedback on this manuscript. The Cell Imaging and Cytometry Core facility (Turku Bioscience, University of Turku, Åbo Akademi University, and Biocenter Finland) and Turku Bioimaging are acknowledged for services, instrumentation, and expertise.

## Author Contributions

**Conceptualization:** Guillaume Jacquemet.

**Data curation:** Guillaume Jacquemet.

**Formal analysis:** Joanna W. Pylvänäinen, Guillaume Jacquemet.

**Funding acquisition:** Guillaume Jacquemet.

**Investigation:** Hanna Grobe, Joanna W. Pylvänäinen, Ricardo Henriques, Guillaume Jacquemet.

**Methodology:** Estibaliz Gómez-de-Mariscal, Guillaume Jacquemet.

**Project administration:** Guillaume Jacquemet.

**Resources:** Guillaume Jacquemet.

**Software:** Estibaliz Gómez-de-Mariscal, Hanna Grobe, Joanna W. Pylvänäinen, Laura Xénard, Jean-Yves Tinevez, Guillaume Jacquemet.

**Supervision:** Ricardo Henriques, Jean-Yves Tinevez, Guillaume Jacquemet.

**Validation:** Hanna Grobe, Joanna W. Pylvänäinen, Guillaume Jacquemet.

**Visualization:** Guillaume Jacquemet.

**Writing – original draft:** Guillaume Jacquemet.

**Writing – review & editing:** Estibaliz Gómez-de-Mariscal, Hanna Grobe, Joanna W. Pylvänäinen, Laura Xénard, Ricardo Henriques, Jean-Yves Tinevez, Guillaume Jacquemet.

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
