## [Editor Report · Decision Letter 0]

6 Feb 2024

Dear Guillaume, 

Thank you for submitting the revised version of your manuscript entitled "CellTracksColab—A platform for compiling, analyzing, and exploring tracking data" to PLOS Biology and for reformatting as a Methods and Resources Article.

Your manuscript has now been evaluated by the PLOS Biology editorial staff, as well as by an academic editor with relevant expertise, and I am writing to let you know that we would like to send your submission out for external peer review.

Once your full submission is complete, your paper will undergo a series of checks in preparation for peer review. After your manuscript has passed the checks it will be sent out for review. To provide the metadata for your submission, please Login to Editorial Manager (https://www.editorialmanager.com/pbiology) within two working days, i.e. by Feb 08 2024 11:59PM.

Best wishes,

Richard

Richard Hodge, PhD

rhodge@plos.org

PLOS

---

## [Decision Letter · Decision Letter 1]

18 Mar 2024

Dear Guillaume,

Thank you for your patience while your manuscript "CellTracksColab—A platform for compiling, analyzing, and exploring tracking data" was peer-reviewed at PLOS Biology as a Methods and Resources Article. Please accept my apologies for the delays that you have experienced during the peer review process. Your manuscript has now been evaluated by the PLOS Biology editors, an Academic Editor with relevant expertise, and by four independent reviewers. Please note that Reviewer #2 has also provided an attachment along with their review that I have provided as part of this letter.

In light of the reviews, which you will find at the end of this email, we would like to invite you to revise the work to thoroughly address the reviewers' reports.

As you will see, the reviewers are generally positive about your tracking tool and think it has the potential to be very useful for the community and non-programmers. However, they provide several overlapping suggestions to improve the presentation/organization and accessibility of the tool. In addition, Reviewer #1 raises concerns with the flexibility of the tool since it only takes TrackMate inputs and with the reliance on Google resources. After discussions with the Academic Editor, we agree that a revised version should include a demonstration that CellTracksColab can be run locally, as well as addressing the limitations regarding the use of Google infrastructure. 

Given the extent of revision needed, we cannot make a decision about publication until we have seen the revised manuscript and your response to the reviewers' comments. Your revised manuscript is likely to be sent for further evaluation by all or a subset of the reviewers.

**IMPORTANT - SUBMITTING YOUR REVISION**

*Re-submission Checklist*

*Published Peer Review*

*PLOS Data Policy*

*Blot and Gel Data Policy*

Sincerely,

Richard

Richard Hodge, PhD

rhodge@plos.org

REVIEWS:

Reviewer #1: Jacquemet presents a tool - CellTracksColab - that relies on Google resources to analyze biological tracking data. I like the idea behind the platform: a simple way to give non-programmers access to advanced analysis. Personally, I would not use it nor encourage my lab to do so (we would need more flexibility and the Google tie-in is a killer), but I can see the utility of the platform. From a publication point of view, there is not so much innovation in the analysis, but that isn't the point. The point is to bring currently available tools to the masses and the platform can definitely meet this goal. I like the way the platform is presented, allowing users to go deeper with successive routines if they want. I don't imaging many novices will want a t-SNE plot of their tracks, but it's there if they do.

The USP is using Google resources, but this is a serious limitation in my view: a) the user needs sufficient privileges to do their analyses (the free tier is restricted), b) there is an issue over data privacy, and even needing a Google account is a stumbling block for anyone concerned about their privacy, c) Google is notorious for shuttering useful platforms - this could all disappear tomorrow, d) uploading massive datasets to do analysis is an accessibility issue. To be fair, the author acknowledges a and b in the manuscript, and offers "To address this challenge, we plan to enable CellTracksColab notebooks to function seamlessly across different platforms via automatically generated Docker images". To my mind, this step is required for publication. This shouldn't all hang off Google and the ability to run this locally, means points a to d are nullified. I would not advocate a Docker image as there are a separate set of issues with that, but it would be straightforward for the author to generate a conda/yml install for people to just run all this stuff in a Jupyter notebook on their machine. If they did this, it would increase the use of the platform. I would be happy to use a local version, but not the online one. Speaking from the experience of using ColabFold, for lightweight predictions performed occasionally, it's OK; for anything even vaguely serious, you need a local install of AlphaFold. While that requires a lot of computing resources, what is being used here is a simple means to run a bit of python code (there isn't a strong case to use Google computing resources for these analyses) locally, and I hope the author agrees this is worth doing.

The second major limitation is that while it is billed as a "do everything" platform, it only takes TrackMate output. If the tracking is done using other software, the outputs would require transforming into TrackMate format. This isn't so clear from my reading of the paper. The other limitation is the use of 3 csv files per tracked experiment. It would be more efficient for users to save the TrackMate XML file and for CellTracksColab to parse that, rather than deal with multiple files. There would be less user error here too. Does the platform currently check that all files are present? Finally, it's on the user to ensure the scaling information of their tracking data is correct. Users, particularly novices, tend to lose the scaling information when they process images. It would be useful if there could be a sanity check implemented. A simple check is if the spatial units are pixels, then warn the user that their results are not scaled. This is harder to do from the CSVs and would be trivial if the platform used TrackMate XML instead. If the author could acknowledge and/or address these limitations, it would improve the platform and manuscript.

Very specific points:

1. The heatmap for Cohen's d which is [0,1] works well. The heatmap for p-values, not so much. First, the colorscale needs to be log transformed for this to be useful. It is also not possible to have p-values that equal 0 or 1, this simply means the numeric precision of the labels used as overlays in the heatmap are insufficient. What would work better here is to not show labels that are ~ 1 (or > 0.1 or some other cut off), then show the -log10 transform of the p-value. So that 1 x 10^-6 can be discerned from 1 x 10^-5.

2. As a follow up, are those p-values shown in Fig 1C calculated correctly? As in, using the means of the repeats (n = 3 for each, superplot style) or using all the tracks (n > 1000s, pseudoreplication) 

3. the link on GitHub "Data Structure: Organize with our two-tiered folder hierarchy" is currently broken

4. The implementation of Ripley's L function shown in Figure 1, how does it deal with search areas that go beyond the image? For low density images like the one shown in 1F this is a significant issue with the calculation.

5. Kudos to the author for making some test datasets available. My point above about the XML files, means that the tracking can be reproduced because the parameters for TrackMate are stored in the XML file, which is not the case for csv. You can tell I am not a fan of the csv files!

6. The author makes the case for analysis of all kinds of tracking data from molecules to whole animals. Have they tested whether CellTracksColab can handle all different kinds of data? Maybe if users report problems tracking objects different to cancer cells or filopodia, they can be patched on-the-fly? It would be good to know if there has been some stress testing done, because the initial parameters offered by CellTracksColab are unlikely to work for every case.

Reviewer #2: Please see attachment, which includes the review along with figures highlighting some minor issues with the code.

The authors of CellTracksColab present an interesting platform for analysing image tracking data, which appears very useful for the community. However, some points should be addressed to make the suggested tool more accessible. 

One question is regarding how some of their metrics are calculated within the platform and which parameters can be tuned. For example, the smoothing option is described in the notebook as "Smoothing: Adjust this slider to control the degree of smoothing you'd like to apply to your tracks." - what is the slider defining? The temporal interval of a moving average? It is possible to retrieve this information by expanding the code cells in the notebook, but this is only true for experienced (coding) users. 

From our understanding, the algorithm calculates speed and persistence of the cell tracks by using the whole length of the track. This represents a limitation, as track speed and persistence are prone to discrepancies based on track length. If one sample has longer tracks than another then this could lead to artificial differences. In the current setup, short and long tracks are grouped together, and their output considered equivalent. An alternative approach would be to run speed and persistence calculations on a pre-defined time window deltaT for each identified track in a moving average manner. The definition of such parameter from the user would be a welcome addition to the notebook.

Other filtering options would also be useful in specific biological situations, such as filtering out cells that are not moving (short trajectories).

The notebook is meant to be novice-oriented. While this stands true thanks to the easy installation and the code being "hidden" from view, the tool can become quite overwhelming for novices. One way around would be to separate the current content into different notebooks (one for data visualisation, one for standard analysis, one for quality control and one for advanced analysis and clustering). This would also allow additional space for lengthier explanations where needed (e.g., smoothing, see above), and would prevent user being put off by advanced features they might not be interested with to start with.

Moreover, while it's very useful for experienced users to be able to access all the computed metrics, for novices this might be discouraging (Fig 1). One suggestion would be to group them (e.g., spatial metrics: directionality, tortuosity, …; shape metrics: area, perimeter, …) and/or to divide them into basic and advanced measurements, with only the basic ones being displayed to start with. A select/unselect all option would also be a welcome addition. 

Figure 1 - a subset of the metrics that can be selected.

While testing the notebook, a few issues were found, which are listed below.

* In Part 1.3 we could not get to work the "Folder_path" input. We initially downloaded the test dataset separately but could not get the notebook to work when setting either the path to the parent folder, or any of the subfolders in the first input row. Everything works smoothly when choosing "Use_test_dataset".

* Despite setting up the Results folder in the same cell, no output was saved at any point. Confirmation messages of saving were displayed at different point in the notebook (no error messages), but the folder remained empty.

* In Part 1.5 it is unclear whether the filtering is happening on one movie or on the whole dataset - we assumed it was on the whole dataset, but further clarifications would be helpful. Additional filters would also be welcome (see text above, e.g., filter by distance travelled, deltaT over which analysis should be performed, further details for smoothing)

* In Part 2, it was unclear at first where the generated output is saved. We acknowledge than this might be related to the problem with output saving reported above, but should it not be the case, we think it would be helpful to save the output in csv format for further analysis and perform some additional visualisations at this stage. This would be particularly relevant if splitting the notebook as suggested above.

* An error occurred in different sections when trying to plot selected metrics (section 4.1, 4.2.3, 5.7.1, 6.7), despite selecting valid metric calculated in the previous cells:

* In Part 4.1 the heatmap returns only ±1 values. Is this an expected behaviour? If yes, why would this be the case?

Reviewer #3: Overall, the author provides a useful collection of track analysis tools running on a widely accessible platform. Many different biological use cases generate track data, and so an analysis package has a potentially wide audience. The tools that enable analysis of track heterogeneity will be particularly welcomed by the community.

Although the tool itself has many potential users, especially non-programmers, it requires that track data be inputted in a very specific way. The author writes "CellTracksColab processes tracking data stored in CSV files, where each file corresponds to the tracking results from an individual video. These files must be organized and named according to the notebook's specifications for optimal functionality." It's unclear whether a non-programmer could feasibly structure the tracking data in this way. A list of software packages that can generate tracks from raw data might also benefit some users. 

As argued by the author, the hierarchical clustering does indeed seem to provide clues to the experimenter regarding outliers. Without a sense of statistical significance or robustness, however, it's difficult to use as a scientific measure to compare populations. Are there ways for the user to assess the stability of the clustering?

There seem to be two main rationales for the HDBSCAN clustering of UMAP data, to find actual clusters and to break the data up into categories for fingerprinting. The different rationales aren't clearly described to the reader though, which is especially confusing given that several example datasets don't appear to have natural clusters. Also, similar to the hierarchical clustering, it's not clear how the reader should interpret the fingerprints, especially when more readily interpretable measures are also included in the package. 

The scatterplots in figures 3 and 4 are too dense to be interpretable. Are there automatic ways that users could avoid generating such plots?

The machine learning for each dataset employs many track metrics that are listed, but not mathematically defined. Where are these metrics defined?

Reviewer #4: Overall impression:

The author presents the tool 'CellTracksColab' for analyzing and visualizing tracking data and also performing downstream statistical analyses and visualization. The described tool appears to be multifunctional and promising and the manuscript is well-written and well-illustrated.

Specific points gathered from the manuscript and while trying out the tool hands-on that speak to the strengths of the tool:

-The authors mention that more than 50,000 tracks, amounting to a staggering ~3 million objects were analyzed using CellTracksColab

-The tool enables automated tracking from videos

-The tool leverages the strength of Python programming language and runs on Google Colab, and thereby makes tracking analysis more accessible for uses with limited programming experience

-Quality control and version control steps are built into the analysis workflow

-While performing track filtering and smoothing, the tool allows side-by-side visualization of 'before' and 'after' graphs, which is helpful

-The graphs present in the manuscript can be generated (and thereby checked) by running the test dataset

Potential areas of improvement:

-Some sections of the notebooks (mentioned below) could be improved / could be better documented to help novice image analysts. Video tutorials could also be helpful in this regard.

-It is a bit hard to understand how the tool is performing quality control

-I was not successful in running the analysis by choosing 'raw data' and not the 'filtered and smoothed data' for the test dataset using the CellTracksColab TrackMate notebook.

-While trying to plot the balanced dataset, the error message 'no parameters are selected' is often received, even when parameters have been set.

-Runtime speeds provided by Google Colab (as the author also mentions in the manuscript) can be a challenge

-The author recommends arranging data into a specific folder hierarchy comprising conditions and repeats. For users with a smaller dataset (e.g. without multiple conditions and/or repeats), it might be helpful to not make this hierarchy mandatory for using the tool or supporting a flat architecture instead.

---

## [Decision Letter · Decision Letter 2]

25 Jun 2024

Dear Dr Jacquemet,

Thank you for your patience while we considered your revised manuscript "CellTracksColab—A platform for compiling, analyzing, and exploring tracking data" for publication as a Methods and Resources at PLOS Biology. This revised version of your manuscript has been evaluated by the PLOS Biology editors, the Academic Editor and the original reviewers.

Based on the reviews, we are likely to accept this manuscript for publication, provided you satisfactorily address the following data and other policy-related requests.

A) Please change your title to: "CellTracksColab is a platform that enables compilation, analysis and exploration of cell tracking data" to follow house style.

B) Please adjust your financial disclosure statement. Ensure that numbers are provided for all grants. It would improve readability if you list all grants first, and end with the statement that funders were not involved in the design and execution of this study. In addition, please move the acknowledgment to the appropriate section

C) Please can you modify the abstract in such a way that it is clear you are tracking cells? We would suggest to modify one of the sentences in the following manner: "Here, we introduce CellTracksColab, a platform tailored to simplify the exploration and analysis of tracking data and demonstrate its capabilities in three different cell lines".

D) DATA POLICY

1C, 2BGH, 3CEGH, 4CD and S4DEF

E) CODE POLICY

We have noted that you have deposited your code on Zenodo, and provided us with the details. However, the CellTracksColab dataframes and the code for CellTracksColab (10.5281/zenodo.10564928) are currently not available. Please ensure these are publicly available.

We expect to receive your revised manuscript within two weeks. 

*Published Peer Review History*

*Press*

Sincerely,

Suzanne

Suzanne De Bruijn, PhD, 

Associate Editor

sbruijn@plos.org

PLOS Biology

Reviewer remarks:

Reviewer #1: The authors have addressed all of my comments in full. I'd like to thank them for their efforts to tackle the issues I raised and I think the platform and paper are much better as a result.

Reviewer #3: The authors addressed all of my comments well. They especially now provide multiple documented ways for users to input tracking data, which will expand their audience. They created software that many people will hopefully soon use.

---

## [Editor Report · Decision Letter 3]

8 Jul 2024

Dear Dr Jacquemet,

Thank you for the submission of your revised Methods and Resources "CellTracksColab is a platform that enables compilation, analysis and exploration of cell tracking data" for publication in PLOS Biology. On behalf of my colleagues and the Academic Editor, Carole Parent, I am pleased to say that we can in principle accept your manuscript for publication, provided you address any remaining formatting and reporting issues. These will be detailed in an email you should receive within 2-3 business days from our colleagues in the journal operations team; no action is required from you until then. Please note that we will not be able to formally accept your manuscript and schedule it for publication until you have completed any requested changes.

Although we agree your updated financial disclosure statement is much improved, we would kindly ask you to include links to the funding agencies in this statement.

PRESS

Sincerely, 

Suzanne 

Suzanne De Bruijn, PhD, 

Associate Editor

PLOS Biology

sbruijn@plos.org